# An anti-occlusion optimization algorithm for multiple pedestrian tracking

**Lijuan Zhang**[1,2], **Gongcheng Ding**[2], **Guanhang Li**[2], **Yutong Jiang**[3]*, **Zhiyi Li**[4], **Dongming Li**[1]

**1** College of Internet of Things Engineering, Wuxi University, Wuxi, China, **2** College of Computer Science and Engineering, Changchun University of Technology, Changchun, China, **3** China North Vehicle Research Institute, Beijing, China, **4** College of Instrument Science and Electrical Engineering, Jilin University, Changchun, China

* jiangyutong201@163.com

## Abstract

Frequent occlusion of tracking targets leads to poor performance of tracking algorithms. A common practice in multi-target tracking algorithms is to re-identify the occluded tracking targets, which increases the number of identity switching occurrences. This paper focuses on online multi-object tracking and designs an anti-occlusion, robust association strategy, and feature extraction model. Specifically, the least squares algorithm and the Kalman filter are used to predict the trajectory of the tracking target, while the two-way self-attention mechanism is employed to extract the features of the tracking target, as well as positive and negative samples. After the tracking target is occluded, the association strategy is used to assign the identity information from before the occlusion. The experimental results demonstrate that the algorithm proposed in this paper has achieved excellent tracking performance on the MOT dataset.

**Data Availability Statement:** The experimental data can be found on the official page of MOT Challenge, https://motchallenge.net/method/MOT=6764&chl=5.

## 1. Introduction

Multi-object tracking, as a major branch in the field of computer vision, has been rapidly developing in recent years and has significant application value in various fields such as intelligent monitoring, action and behavior analysis, autonomous driving, virtual reality, and more.

The field of multi-object tracking is mainly divided into two categories: offline tracking algorithms, such as MHT, Interacting Tracklets for Multi-Object Tracking, learning a neural solver for multiple object tracking [1–3], and online tracking algorithms, such as AS2RCF, DCOT, Tracktor++, CenterTrack, FairMot, and TransMOT [4–9]. As offline tracking algorithms cannot be applied to real-time tracking, online tracking algorithms are the current trend in multi-object tracking research.

Bewley et al. [10] proposed a simple online and real-time tracking algorithm, SORT (Simple Online and Realtime Tracking), which cleverly decomposes the multi-object tracking problem into three parts: the object detection part responsible for providing target bounding boxes, the state prediction part responsible for predicting and updating trajectory information, and the data association part responsible for solving the matching problem between targets and

**Funding:** The author(s) received no specific funding for this work.

**Competing interests:** The authors have declared that no competing interests exist.

trajectories. The input image is processed by an object detection algorithm to output the position and category of each detected target, which is then predicted and updated using Kalman filtering. Finally, the Hungarian algorithm is used to solve the cost matrix of the predicted targets and the detected targets in the current frame with IOU matching. The SORT algorithm has fast tracking speed but does not solve the problem of frequent identity switching. JDE (Jointly learns the Detector and Embedding model) [11] detects objects in the image using a detector and matches objects across frames based on the appearance features and motion patterns of objects in the detection boxes. This achieves multi-object tracking and is the first multi-object tracking algorithm to approach real-time. The algorithm has high accuracy but results in frequent identity switching.

FairMOT addresses the problem of Anchor-based detectors not being compatible with JDE tracking mode by using the Anchor-Free [12] object detection paradigm instead, which estimates the center of the object on a high-resolution feature map. To address the problem of multi-level feature fusion, the DLA (Deep Layer Aggregation) [13] network is selected to improve the ResNet-34 [14] backbone network for multi-level fusion feature extraction, processing targets of different scales and achieving excellent tracking performance. This effectively reduces the number of identity switches. Zhang et al. [15] proposed a simple and efficient data association method called BYTE, which used the similarity between the detection boxes and tracking trajectories to retain high-scoring detection results while removing the background from low-scoring detection results, thereby uncovering true targets (such as difficult samples due to occlusion or blur) and reducing missed detections while improving the coherence of trajectories. This approach achieves a lower number of identity switches on the MOT17 and MOT20 [16] datasets. Zhang et al. [17] combined deep features with handcrafted features and proposed a new Robustness Criterion for evaluation. They used an adaptive threshold to determine whether to use the Siamese network for re-detection, effectively addressing occlusion and background clutter issues in object tracking and achieving state-of-the-art performance. Similar object interference in object tracking often leads to tracking drift, Huang et al. [18] introduced a compensated attention model, which incorporated attention mechanisms in the feature extraction modules of both the template branch and search branch of the Siamese network. This model enhances the feature representation of both the target and the similar backgrounds simultaneously and improves the discriminative ability of the search branch towards the object. Target tracking requires establishing the relationship between objects in the previous and the current image. Using a detection algorithm in a single frame can only achieve recognition but not data association. Using a single-object tracking algorithm between adjacent images can only achieve the association of a single object but not recognition. The replacement of old and new objects and identity detection are major challenges in multiple object tracking (MOT). The detection model and association strategy affect the performance of MOT tracking algorithms. The DeepSort [19] algorithm introduces the Reid (Re-identification) feature extraction model to extract target feature information and uses cosine distance to calculate the similarity of target feature information. The algorithm also uses Mahalanobis distance to constrain target motion information for data association, effectively reducing the number of identity switches. It has achieved good results on the MOT16 [20] dataset.

In this paper, we first propose a dual-path self-attention mechanism model, which uses a cyclic shift to construct a large number of target negative samples for training. The self-attention mechanism is used to extract global features of the targets, thereby improving the model's robustness and feature extraction ability. To address the problem of frequent target identity switching in existing multi-object tracking algorithms due to occlusion, we propose a new association strategy. Different algorithms are used to predict the motion trajectory of targets that are frequently occluded and targets that are occluded for a long time. Some detection

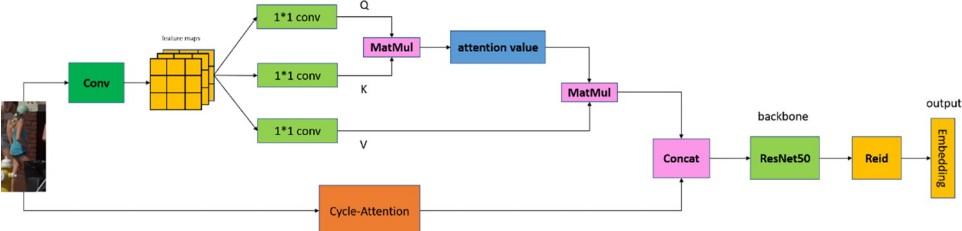

**Fig 1. Feature extraction model diagram.** Images republished from Vidsplay.com [21] under a CC BY license, with permission, original copyright [2023].

boxes are set as high-value detection boxes, which are used to associate the targets that have just emerged from occlusion effectively reducing the number of identity switches during tracking.

The experimental results show that our optimized algorithm in this paper achieves excellent performance on the MOT dataset. Specifically, our proposed tracking algorithm performs significantly well under high resolution and high object detection confidence conditions.

## 2. Network model design based on self-attention mechanism

In this paper, we design a new dual-path attention feature extraction module, as shown in Fig 1.

### 2.1 Fusion of cyclic shift and self-attention mechanism

The target detection boxes in dense multi-object detection often contain a large amount of background area, and there are many small targets, making the commonly used feature extraction models less effective. In this paper, a large number of negative samples are constructed by cyclically shifting the tracking targets for training. Each negative sample has the same upper and lower bounds in each channel, and it can be assumed that each color component of any randomly selected pixel is independently and identically distributed. The Attention mechanism [22] is used to select a small amount of important information from a large amount of information, and the global association weights are used to perform the weighted sum for the input, reducing dependence on external information and focusing on capturing internal correlations in the data or features. After adding the feature information of the positive samples at the channel level and using the self-attention mechanism, the model can converge to the optimal solution faster. Finally, ResNet50 is used as the backbone for training, and the Cycle-Attention module is shown in Fig 2. In this paper, the two-dimensional image is reduced in dimension, and the number of cyclic shifts is 1/16, 1/8, and 1/4 of the original image.

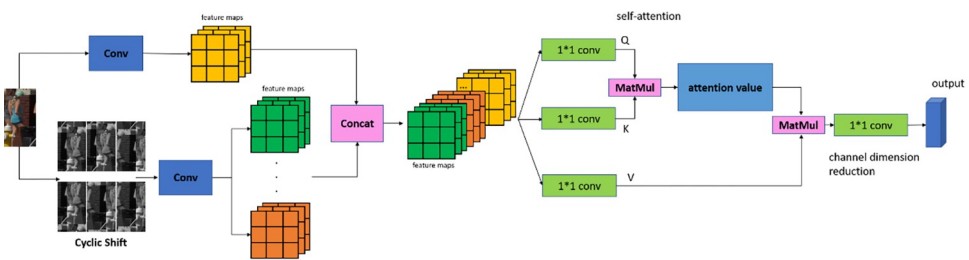

**Fig 2. Cycle-attention module.** Images republished from Vidsplay.com [21] under a CC BY license, with permission, original copyright [2023].

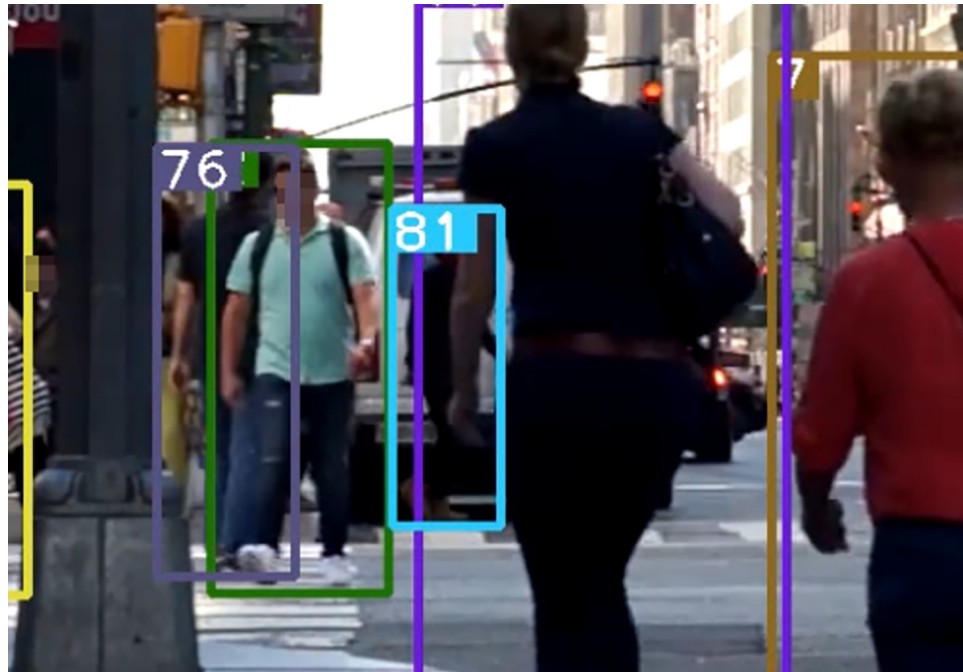

**Fig 3. Prediction box matching (81).** Images republished from Vidsplay.com [21] under a CC BY license, with permission, original copyright [2023].

The network feature extraction module proposed in this paper that enhances the model's generalization ability and robustness, enabling better handling of interference caused by background and motion. With the application of our proposed model, the tracking algorithm's performance has significantly improved, as demonstrated in the ablation experiments in **Section 4.1**.

## 3. High value prediction box target association strategy

We propose a more robust association strategy, which selects an appropriate trajectory prediction algorithm based on the number of lost frames caused by the occlusion of the target detection box. The high-value prediction boxes are used for association, effectively reducing the number of identity switches during the tracking process.

### 3.1 High value prediction box selection and matching

When common multiple object tracking algorithms such as DeepSORT perform target tracking tasks, detection boxes in the non-deterministic state will be discarded (the non-deterministic state refers to the box not being detected in three consecutive frames). In Fig 3, the predicted box is successfully matched, but the 81th detection box did not enter the deterministic state. In Fig 4, the predicted box is mismatched, and in Fig 5, the lack of appropriate detection boxes for association leads to identity switching.

The discarded non-determined detection boxes still have association value and are retained in the association strategy proposed in this paper. Meanwhile, the detection boxes that are deleted due to exceeding the maximum lifespan are also retained. This type of the detection boxes also has association value for long-term target occlusion. Typically, most long-term occluded targets lose suitable matching boxes due to the detection box exceeding the

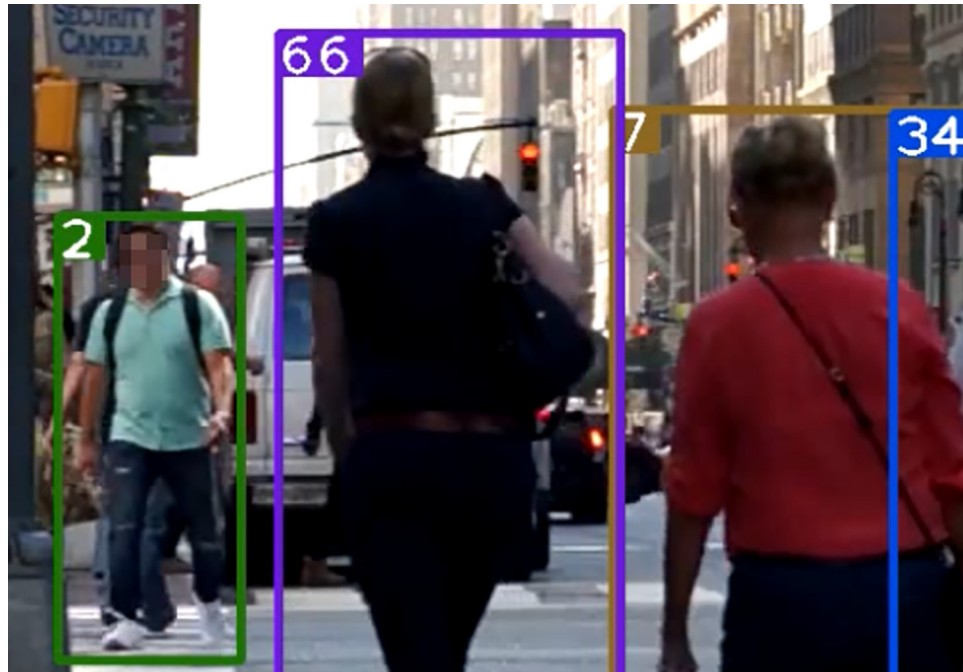

**Fig 4. Prediction box mismatch.** Images republished from Vidsplay.com [21] under a CC BY license, with permission, original copyright [2023].

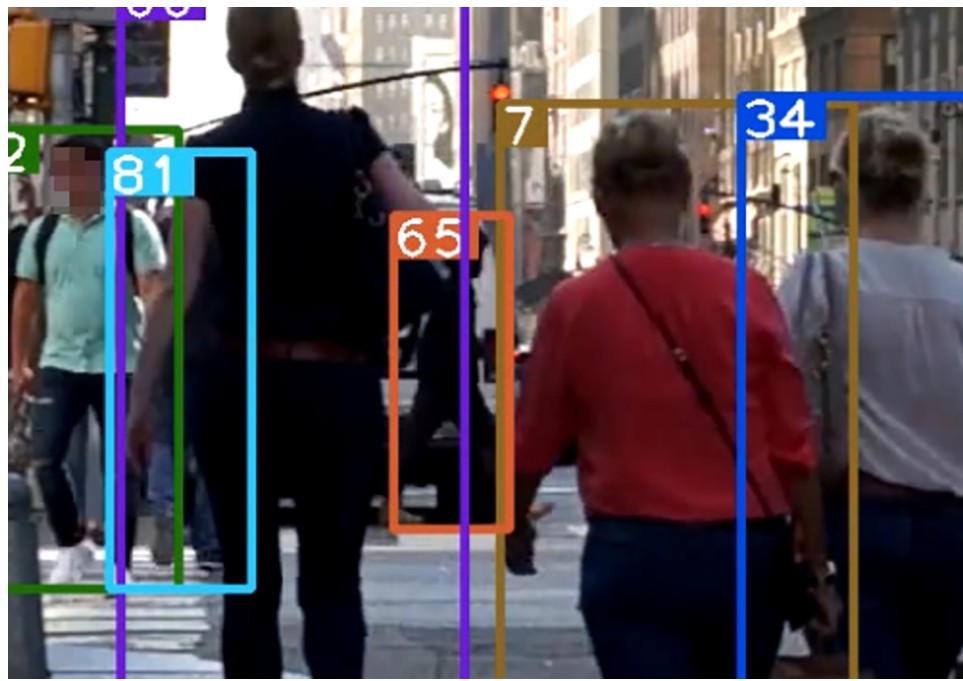

**Fig 5. Prediction box identity switching (65).** Images republished from Vidsplay.com [21] under a CC BY license, with permission, original copyright [2023].

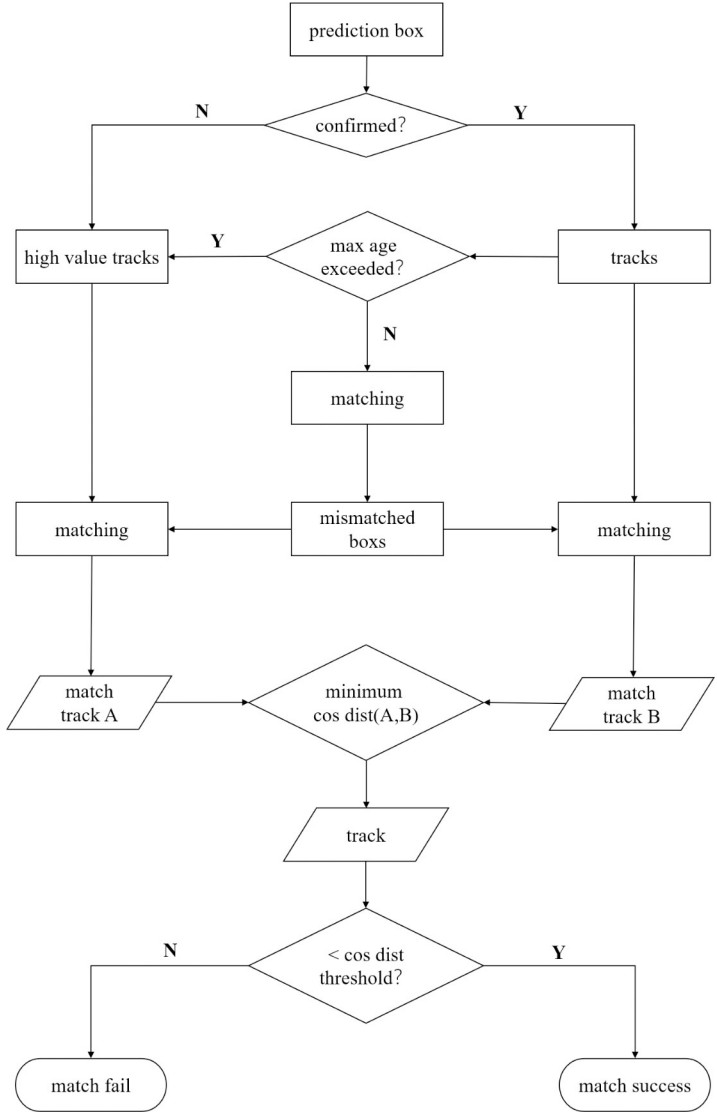

**Fig 6. Matching flow chart.**

maximum lifespan, which leads to association failure. We set these two types of detection boxes as high-value detection boxes and associates them after target occlusion in this paper. The association process is shown in Fig 6.

When performing association in a certain frame, the mismatched detection boxes are associated with both the high-value detection boxes and the predicted boxes. The feature information is extracted by our proposed network module, and the corresponding cosine distance is calculated. The one with the smallest cosine distance is the most suitable association item. Moreover, the high-value detection boxes are only used once when associated to prevent frequent switching caused by different tracking targets competing for high-value detection boxes in a short video frame.

After using the association strategy described in this paper, Fig 7 shows a successfully matched predicted bounding box and an undetermined detection box with ID 44 retained. Fig 8 shows a mismatched predicted bounding box, and Fig 9 shows a successfully matched

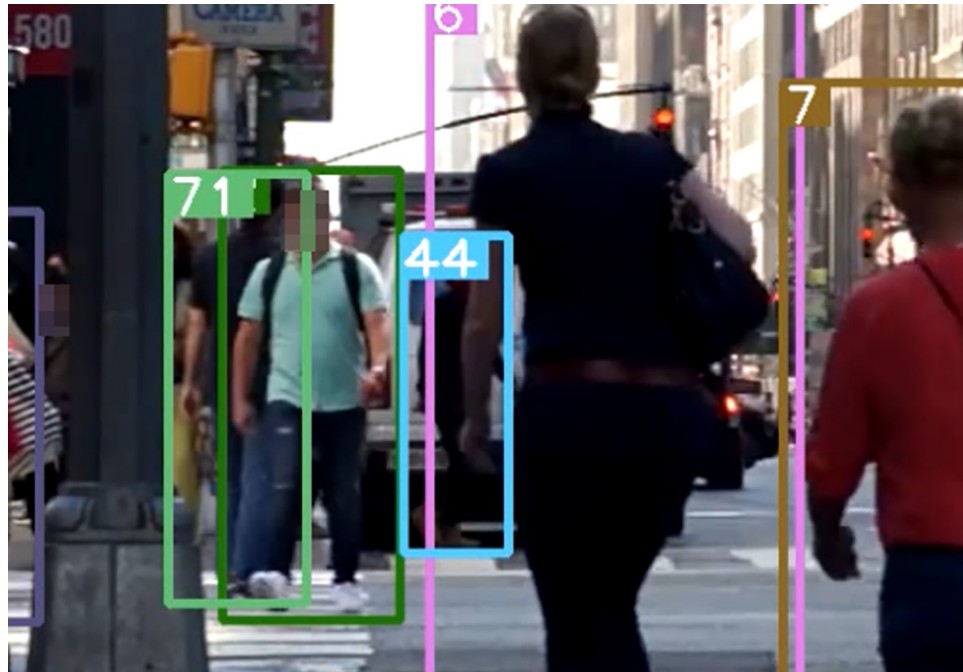

**Fig 7. Prediction box matching (44).** Images republished from Vidsplay.com [21] under a CC BY license, with permission, original copyright [2023].

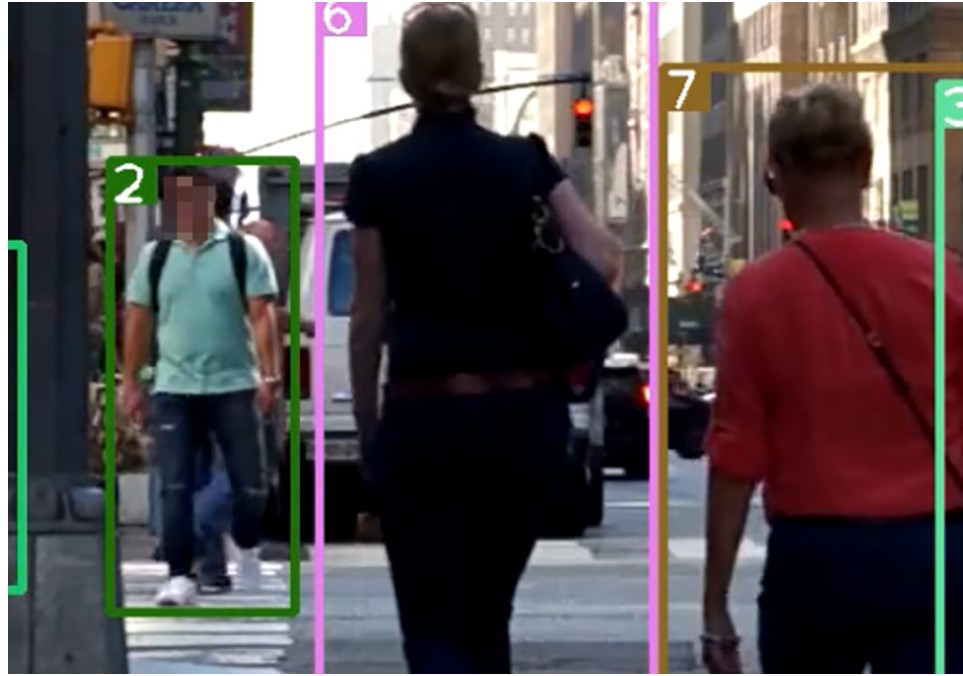

**Fig 8. Prediction box mismatch.** Images republished from Vidsplay.com [21] under a CC BY license, with permission, original copyright [2023].

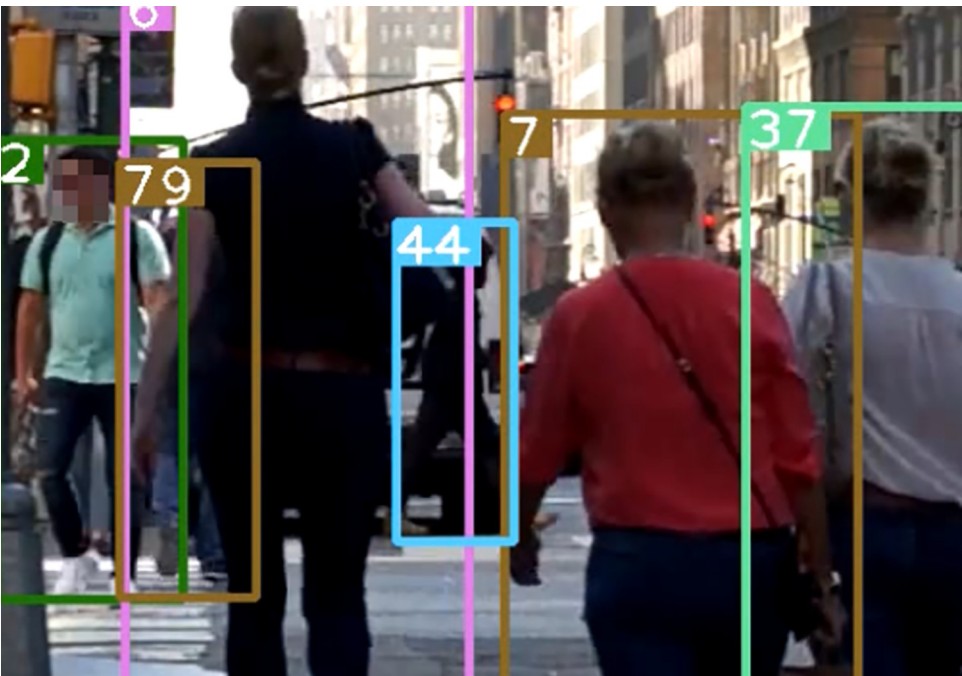

**Fig 9. Prediction box identity switching (44).** Images republished from Vidsplay.com [21] under a CC BY license, with permission, original copyright [2023].

predicted bounding box with the high-value predicted box with ID 44 successfully associated with a previously mismatched detection box after the tracked target becomes no occluded blocks.

## 3.2 Introducing the least square method to reduce ID switching under frequent occlusion

When tracking targets that are occluded, algorithms such as DeepSORT and SORT use Kalman filters to predict trajectories. However, when targets are frequently occluded, there are fewer actual measurement data available, which can result in inaccurate Kalman gain calculations and large deviations between the estimated and true states of the Kalman filter. This can lead to large prediction errors in the trajectory and failed association between the detection box and the tracked target after the occlusion ends, causing identity switches. In this paper, we introduce the least squares algorithm to fit the target's motion trajectory in response to frequent occlusions. Using fewer video frames, our algorithm can effectively predict the target's motion trajectory.

The target's motion trajectory can be considered linear during short-term intervals, and the least squares method finds the best function match for the data by minimizing the squared error. During the target tracking process, the center point coordinates $(x_i, y_i)$ of the target tracking box are retained for each frame. When the target experiences a short-term occlusion, the number of center point coordinates equals the number of motion trajectory information, denoted as $n$.

The least squares error function is constructed as shown in Eq (1).

$$S = \sum_{i=1}^{n} [y_i - (ax_i + b)]^2 \tag{1}$$

In the equation, $S$ represents the fitting error; $n$ represents the number of motion trajectory information; $x_i$, $y_i$ represent the trajectory information, that is, the center point coordinates; $a$, $b$ represent the slope and intercept of the trajectory.

$$\frac{\partial}{\partial a}S^2 = \frac{\partial}{\partial a}\sum_{i=1}^{n}[y_i - (ax_i + b)]^2 = 0 \tag{2}$$

$$\frac{\partial}{\partial b}S^2 = \frac{\partial}{\partial b}\sum_{i=1}^{n}[y_i - (ax_i + b)]^2 = 0 \tag{3}$$

$$(f(x) \pm g(x))' = f'(x) + g'(x) \tag{4}$$

By taking the partial derivative of in Eq (2) and Eq (3) and setting it to 0, the minimum value of the extreme point of $S^2$ can be calculated. The trajectory information of the target, $x_i$, $y_i$, is known, so the problem is transformed into solving $a$, $b$, the trajectory parameters.

Taking $a$ as a constant, find the partial derivative of and apply it to Eq (4) to get Eq (5).

$$\left(\sum_{i=1}^{n}x_i\right)*a + n*b = \sum_{i=1}^{n}y_i \tag{5}$$

By using the least squares algorithm to convert known center point coordinates and trajectory information into motion trajectory, when the occlusion of the target ends and the predicted bounding box needs to be matched with a detection box, we use the target matching strategy described in **Section 3.1** for association, as shown in Fig 10.

For long occlusions of a tracked target, there are more actual measurement values available, and the Kalman gain calculation is more accurate. Therefore, we use the Kalman filter for prediction that is more appropriate. Assuming that the dynamic system has multiple random

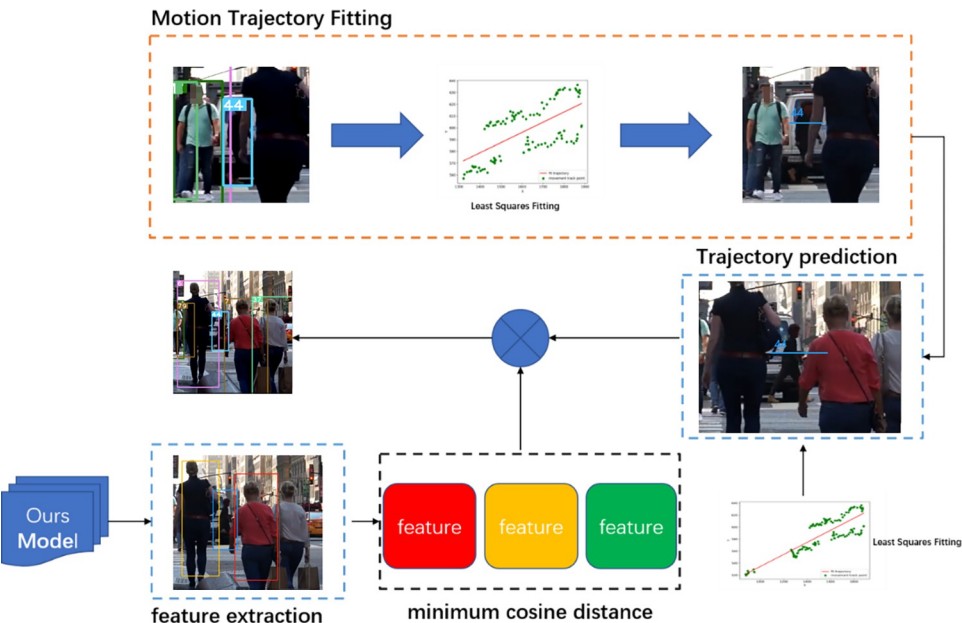

**Fig 10. Least squares handling occlusion.** Images republished from Vidsplay.com [21] under a CC BY license, with permission, original copyright [2023].

variables that follow a Gaussian distribution, and each variable has a mean of $\mu$ and a variance of $\sigma^2$:

1. The covariance matrix measures the correlation between random variables; the value of the matrix represents the degree of correlation between variables.

2. At each time point, there exists the best estimate $\hat{x}$ and covariance matrix $P$ for all variables.

3. There is uncertainty in the dynamic system, and there exist process noise $w$ and measurement noise $v$ at each time point.

$$\hat{X}_k = \frac{1}{k}(z_1 + z_2 + z_3 + \cdots + z_k) \tag{6}$$

$$\hat{X}_k = \hat{X}_{k-1} + \frac{1}{k}(z_k - \hat{X}_{k-1}) \tag{7}$$

$$\hat{X}_k = \hat{X}_{k-1} + K(z_k - \hat{X}_{k-1}) \tag{8}$$

where $\hat{X}$ is the estimated value, $k$ is the time unit, $K$ is the Kalman gain, and $z$ is the measured value.

Assuming the inter-frame displacement follows a linear constant velocity model, $x = [u, v, s, r, \dot{u}, \dot{v}, \dot{s}]^T$, the horizontal and vertical coordinates denoted by $u$ and $v$ respectively, the tracked target center are obtained. The area of the bounding box is represented by $s$, and $r$ represents the aspect ratio of the bounding box. Information on the changes in the state corresponding to the states is introduced to describe the motion state.

By definition of the estimated value, the estimated value at time $k$ in Eq (6) can be transformed from the measured value at time $k$ and the estimated value at time $k$-1, as shown in Eq (7). Eq (8) defines $K$ as $1/k$, which is the Kalman gain. When there are multiple random variables, data fusion is used to solve for $K$.

$$P = \begin{pmatrix} \sigma_x{}^2 & \sigma_x\sigma_y & \sigma_x\sigma_z \\ \sigma_y\sigma_x & \sigma_y{}^2 & \sigma_y\sigma_z \\ \sigma_z\sigma_x & \sigma_z\sigma_y & \sigma_z{}^2 \end{pmatrix} \tag{9}$$

In the equation, $x$, $y$, $z$ are assumed random variables; $\sigma_x{}^2$, $\sigma_y{}^2$, $\sigma_z{}^2$ are the variances of the random variables; $\sigma_x\sigma_y$, $\sigma_x\sigma_z$, $\sigma_y\sigma_z$ are the covariances between the random variables; $P$ is the covariance matrix.

The random variables at time unit $k$ are correlated, and the covariance matrix is constructed based on the assumed random variables and their corresponding variances and covariances. The values in the $P$ matrix indicate the degree of linkage between the variables [23].

When a tracking target is detected, initialize the state vector and its covariance, and use Eq (10) and Eq (11) for state prediction.

$$X_k = AX_{k-1} + Bu_{k-1} + w_{k-1} \tag{10}$$

$$z_k = HX_k + v_k \tag{11}$$

In the equation, $X_k$ represents the system prediction information at time $k$; $z_k$ represents the measurement information at time $k$; $w$ represents the process noise; $v$ represents the measurement noise; $A$, $B$, $H$ are the parameter matrices of the motion equation.

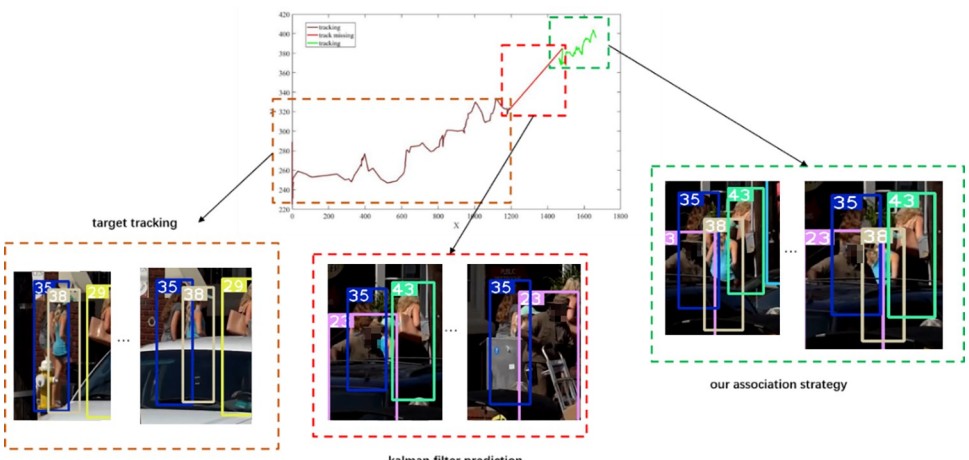

**Fig 11. Kalman filter predicts trajectories.** Images republished from Vidsplay.com [21] under a CC BY license, with permission, original copyright [2023].

The Kalman filter seeks the optimal Kalman gain under the influence of noise, that is, the minimum trace of the covariance matrix of the error $X_k - \hat{X}_k$.

$$K = P_{k-1}H^T(HP_{k-1}H^T + R)^{-1} \tag{12}$$

$$\hat{X}_k = \hat{X}_{k-1} + K(\vec{z}_k - H_k\hat{X}_{k-1}) \tag{13}$$

$$P_k = P_{k-1} - KHP_{k-1} \tag{14}$$

where $R$ represents the noise matrix.

After calculating the Kalman gain $K$, we use Eqs (13) and (14) to update the tracked target state information. When the long occlusion of the target ends, we also use the association strategy described in the paper to associate the detection box with the tracked target, as shown in Fig 11, where the identity information of ID 6 is successfully associated.

### 3.3 Integration of association strategy

The anti-occlusion association strategy proposed in this paper is integrated with the Deepsort algorithm. The implementation process is as follows:

```
Algorithm: anti-occlusion association strategy
Input: unmatched detetctions u_detections; high value tracks miss_-
tracks; all tracks tracks; Euclidean distance threshold edt; scale
threshold st; track missing tims mt; matched detetctions matches;
Output: Matched predicted and detected boxes
Initialization: cos_value; replace_track ← None;
for detection in u_detections do
    for track in miss_tracks do
        /* ed stands for Euclidean distance, and sd stands for scale
information */
        ed ← compute u_detection and track ed
        sd ← compute u_detection and track sd
        if ed < edt and sd < st then
            save track
```

```
        end
    cos_matrix ← compute save track and detecion cosine_distance
    cos_value ← min(cos_matrix)
    replace_track ← track
    end
    for track in tracks do
        /* time_since_update represents the number of frames since
the predicted bounding box disappeared */
        if t. time_since_update < = mt do
            Movement track ← get last suares ftting
        else do
            Movement track ← get Kalman filter
    end
    ed ← compute u_detection and track ed
    sd ← compute u_detection and track sd
    if ed < edt and sd < st then
        save track
    end
    cos_matrix ← compute save track and detecion cosine_distance
    value ← min(cos_matrix)
    if cos_value < value do
        use miss_track
        miss_track remove track
    else do
        use tracks track
    end
    if track not in matches do
        matchs add track
    else do
        return
    end
 end
```

Our proposed association strategy effectively solves the problem of target identity switching caused by short-term and long-term occlusions, as shown in Fig 12. The top-left (375 frames) and top-right (417 frames) differ by 42 frames, while the bottom-left (816 frames) and bottom-right (1017 frames) differ by 201 frames.

The target was occluded for 200 frames, and our proposed algorithm in this paper successfully matched the predicted bounding box with the detected bounding box.

In the ablation experiment (Section 4.1), our proposed association strategy can effectively reduce the number of identity switches. Moreover, in the MOT17 test dataset, compared with other algorithms that use the same detection framework, our proposed algorithm has a lower number of identity switches.

## 4. Experimental results and discussion

We use YOLOv5 as the pedestrian detection framework. The model uses pre-trained weights from the COCO dataset [24], and the feature extraction model is trained on the Market dataset. The pedestrian tracking dataset used is the MOT16 dataset.

### 4.1 Ablation experiment

Tables 1 and 2 show the experimental results for the DeepSort algorithm with YOLOv5, the experimental data by using our proposed optimized model, and the experimental data by

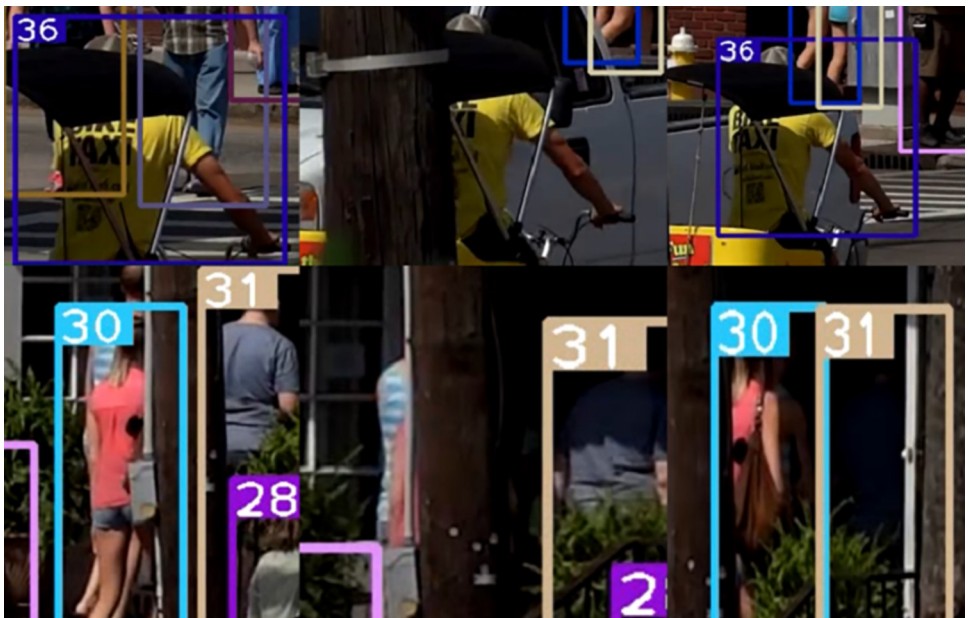

**Fig 12. Algorithmic occlusion processing.** Images republished from Vidsplay.com [21] under a CC BY license, with permission, original copyright [2023].

using our optimized model with our proposed association strategy, with pedestrian confidence scores of 0.3 and 0.5, respectively.

We apply the association strategy and feature extraction model in this paper to MOT15 [26], MOT17, and MOT20 datasets. Experimental results are shown in Tables 3–5, where the detection model confidence threshold is set to 0.3. The MOT17 dataset uses the default DPM [27], Faster-RCNN [28], and SDP [29] three detection inputs.

Due to the low resolution of the MOT15 dataset, the association strategy and feature extraction model in this paper resulted in a decrease in MOTA and HOTA metrics.

The experimental results show that our proposed optimization model and association strategy have improved the MOTA, IDF1, and HOTA metrics. The number of identity switches on the MOT16 dataset was reduced by 30.2% (confidence threshold of 0.3) and 26.3% (confidence threshold of 0.5). On the MOT17 dataset, the total number of identity switches is reduced by 20%. The HOTA index is evaluated through three sub-tasks detection, association and positioning. In Fig 13, our algorithm is compared with the DeepSORT algorithm on the HOTA index. In terms of association accuracy AssA and HOTA score, we have improved the similarity threshold alpha algorithm for different positioning in this paper.

On the MOT20 and MOT15 datasets, the total number of identity switches was reduced by 24.6% and 16.8%, respectively. The detection model of the MOT16 dataset was changed, and experiments were conducted using the DPM [26] detector and CenterNet [30] detector, and the experimental results are shown in Tables 6 and 7.

**Table 1. MOT16 ablation experiment (confidence 0.3).**

| Method | MOTA↑ | MOTP↓ | IDF1↑ | HOTA↑ | IDSW↓ | ML↓ | FN↓ | HZ |
|---|---|---|---|---|---|---|---|---|
| Deepsort [19]+Yolov5 | 46.4 | **77.1** | 53 | 42.9 | 654 | 144 | 60975 | **16** |
| Ours-Model | **46.7** | **77.1** | 53.4 | 43.2 | 537 | 143 | 60516 | 13 |
| Ours-Model+Ours-Association-strategy | **46.7** | **77.1** | **55.5** | **44.1** | **456** | **141** | **58323** | 13 |

**Table 2. MOT16 ablation experiment (confidence 0.5).**

| Method | MOTA↑ | MOTP↓ | IDF1↑ | HOTA↑ | IDSW↓ | ML↓ | FN↓ | HZ |
|---|---|---|---|---|---|---|---|---|
| Deepsort [19]+Yolov5 | 48.5 | 77.8 | 53.5 | 42.9 | 513 | 168 | 62962 | **16** |
| Ours-Model | 48.8 | 77.8 | 54.7 | 43.7 | 422 | 163 | 61774 | 14 |
| Ours-Model+Ours-Association-strategy | **49.1** | **77.7** | **55.8** | **44.6** | **378** | **156** | **60582** | 14 |

The multi-object tracking metrics are as follows:

MOTA: This measure combines three error sources: false positives, missed targets, and identity switches.

MOTP: Accurate location of tracking.

IDF1: The ratio of correctly identified detections over the average number of ground-truth and computed detections.

HOTA [25]: Geometric mean of detection accuracy and association accuracy. Averaged across localization thresholds.

IDSW: Number of Identity Switches (ID switch ratio = #ID switches / recall)

ML: The ratio of ground-truth trajectories that are covered by a track hypothesis for at most 20% of their respective life span.

FN: The total number of false negatives (missed targets).

HZ: Processing speed.

**Table 3. MOT15 ablation experiment (confidence 0.3).**

| Method | MOTA↑ | MOTP↓ | IDF1↑ | HOTA↑ | IDSW↓ | ML↓ | FN↓ | HZ |
|---|---|---|---|---|---|---|---|---|
| Deepsort [19]+Yolov5 | **42.9** | 76.9 | **55.3** | **43.4** | 375 | 89 | 18617 | **31** |
| Ours-Model | 42.7 | 76.9 | 54 | 42.4 | 340 | 87 | 18943 | 28 |
| Ours-Model+Ours-Association-strategy | 42.8 | **76.8** | 54.6 | 42.5 | **312** | **86** | 18635 | 28 |

**Table 4. MOT17 ablation experiment.**

| Method | MOTA↑ | MOTP↓ | IDF1↑ | HOTA↑ | IDSW↓ | ML↓ | FN↓ | HZ |
|---|---|---|---|---|---|---|---|---|
| Deepsort [19] | 72.1 | 81.3 | 69 | 56.8 | 2299 | 185 | 124741 | **6** |
| Ours-Model | 72.4 | 81.3 | **70** | **58** | 1961 | **177** | **120243** | 4 |
| Ours-Model+Ours-Association-strategy | **72.5** | **81.2** | 69.5 | 57.1 | **1841** | 177 | 124311 | 4 |

**Table 5. MOT20 ablation experiment (confidence 0.3).**

| Method | MOTA↑ | MOTP↓ | IDF1↑ | HOTA↑ | IDSW↓ | ML↓ | FN↓ | HZ |
|---|---|---|---|---|---|---|---|---|
| Deepsort [19]+Yolov5 | 15.6 | 73.3 | 18.3 | 14.9 | 3961 | 1564 | 1012823 | **14** |
| Ours-Model | 15.9 | **73.2** | 19.1 | 15.3 | 3426 | 1560 | 1007605 | 11 |
| Ours-Model+Ours-Association-strategy | **16.4** | **73.2** | **20.1** | **15.7** | **2986** | **1546** | **1000388** | 11 |

**Table 6. MOT16 ablation experiment DPM.**

| Method | MOTA↑ | MOTP↓ | IDF1↑ | HOTA↑ | IDSW↓ | ML↓ | FN↓ | HZ |
|---|---|---|---|---|---|---|---|---|
| Deepsort+DPM [19, 27] | 28.4 | 78.4 | 34.2 | 27.8 | 458 | 318 | 85498 | 7 |
| Ours-Model | 28.8 | 78.4 | 35 | 28.3 | 395 | **312** | 84912 | 7 |
| Ours-Model+Ours-Association-strategy | **29.2** | **78.3** | **36.6** | **29.7** | **338** | **312** | **83601** | 7 |

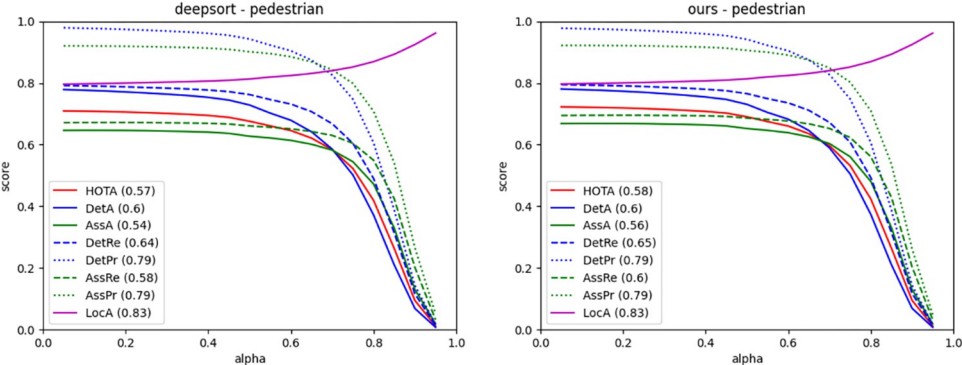

**Fig 13. Comparison results of HOTA indicators.**

**Table 7. MOT16 ablation experiment CenterNet.**

| Method | MOTA↑ | MOTP↓ | IDF1↑ | HOTA↑ | IDSW↓ | ML↓ | FN↓ | HZ |
|---|---|---|---|---|---|---|---|---|
| Deepsort+CenterNet [19, 30] | 54.6 | **80.4** | 60 | 47.7 | 594 | 90 | **52644** | **5** |
| Ours-Model | 58.1 | 81.6 | **62.3** | **49.4** | 488 | 91 | 53392 | **5** |
| Ours-Model+Ours-Association-strategy | **58.5** | 81.6 | 59 | 47.6 | **450** | **88** | 56116 | **5** |

**Table 8. MOT16 test set comparison experiment.**

| Method | MOTA↑ | IDF1↑ | HOTA↑ | IDSW↓ | ML↓ | FN↓ |
|---|---|---|---|---|---|---|
| OUTrack_fm_p [31] | **69.3** | **66.7** | **54.5** | 1298 | **124** | **41177** |
| Lif_TsimInt [32] | 57.5 | 64.1 | 49.6 | **335** | 263 | 72868 |
| MPTC [35] | 62.9 | 65.1 | 51.1 | 685 | 240 | 63565 |
| Tracktor++ [6] | 54.4 | 52.5 | 42.3 | 682 | 280 | 79149 |
| KCF16 [33] | 48.8 | 47.2 | 37.2 | 906 | 289 | 86567 |
| Ours | 64.0 | 62.4 | 51.0 | 838 | **124** | 47849 |

## 4.2 MOT challenge

Conduct experiments on the official datasets MOT16 and MOT17. For the public detection evaluation, we follow the works in OUTrack_fm_p [31] to refine the public detections and keep the bounding boxes that are close to the tracked objects. The performance of the tracking algorithm in this paper can be found in Tables 8 and 9.

## 4.3 Results evaluation

When tracking algorithms such as SORT and DeepSORT use Kalman filters to predict trajectories, frequent occlusions can lead to ID switches in the tracked targets. In this paper, the least squares method and Kalman filter are introduced to handle long-term and short-term occlusions separately, resulting in more reasonable predictions of object tracking bounding boxes.

The average density of the object tracking dataset is relatively high, and frequent occlusions lead to more background clutter. The feature information extracted by the ReID model is often insufficient. In this paper, we design a dual-path self-attention mechanism module that combines cyclic shift operations with self-attention mechanisms. Our module exhibits higher robustness and feature extraction capabilities. In the ablation experiment (Section 4.1), our

**Table 9. MOT17 test set comparison experiment.**

| Method | MOTA↑ | IDF1↑ | HOTA↑ | IDSW↓ | ML↓ | FN↓ |
|---|---|---|---|---|---|---|
| PermaTrack [34] | **73.1** | 67.2 | 54.2 | 3571 | 450 | **123508** |
| OUTrack_fm_p [31] | 69.0 | 66.8 | 55.5 | 3639 | 615 | 140457 |
| Byte_Track [15] | 67.4 | **70** | **56.1** | 1331 | 735 | 172636 |
| MPTC [35] | 62.6 | 65.8 | 51.7 | 4074 | 750 | 198338 |
| Lif_TsimInt [32] | 58.2 | 65.2 | 50.7 | **1022** | 791 | 217944 |
| TADN [36] | 54.6 | 49.0 | 54.8 | 4472 | 464 | 141580 |
| Tracktor++ [6] | 53.5 | 52.3 | 42.1 | 2072 | 861 | 248047 |
| JBNOT [37] | 52.6 | 50.8 | 41.3 | 3050 | 844 | 232659 |
| FAMNet [38] | 52.0 | 48.7 | 0.0 | 3072 | 787 | 253616 |
| Ours | 63.2 | 61.8 | 50.5 | 2616 | **414** | 157581 |

proposed feature extraction model shows improvements in accuracy metrics. The more accurate the object detection model, the higher the accuracy improvement achieved by our model. Common object tracking algorithms usually consider cases where the number of object detection frames is low (during frequent occlusions) or when the tracking boxes have been absent for a long time. In the case of a low number of detection frames, the detection box is not used, while in the case of a long absence of tracking boxes, the detection boxes are often neglected to prevent computational overhead, resulting in increased identity switches of the tracked targets. The correlation strategy designed in this paper considers these types of detection boxes and effectively reduces the number of identity switches during object tracking. As shown in Section 3.3, the high-value detection boxes selected in this paper are used only once during target association, reducing identity switches while preventing false detections and additional costs. In the ablation and comparative experiments, our algorithm achieves a lower ML while reducing the number of identity switches.

**MOT15.** MOT15 consists of 11 sequences in various indoor or outdoor public pedestrian scenes. Both our model and correlation strategy can reduce the number of identity switches. This dataset has a lower resolution, resulting in less pixel information and the loss of some details and crucial information. Our model uses cyclic shifts to construct negative samples, which are suitable for low-resolution targets but are more susceptible to background noise, leading to decreased accuracy. In future work, super-resolution reconstruction or image

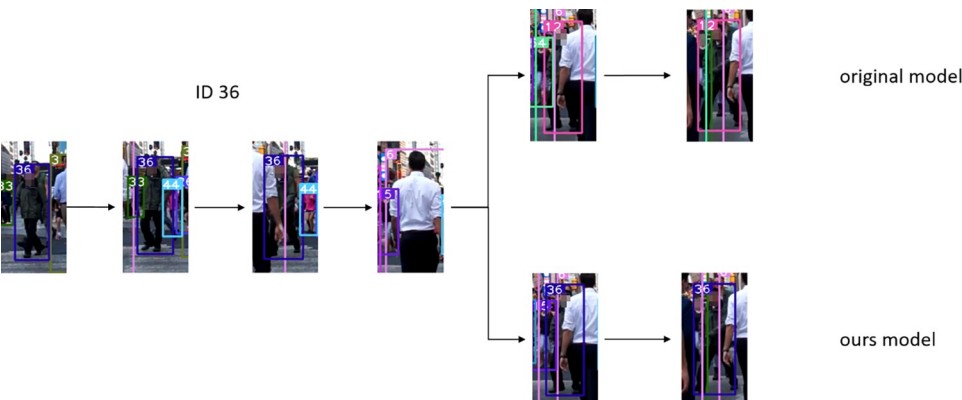

**Fig 14. The result of the target complete occlusion experiment.** Images republished from Vidsplay.com [21] under a CC BY license, with permission, original copyright [2023].

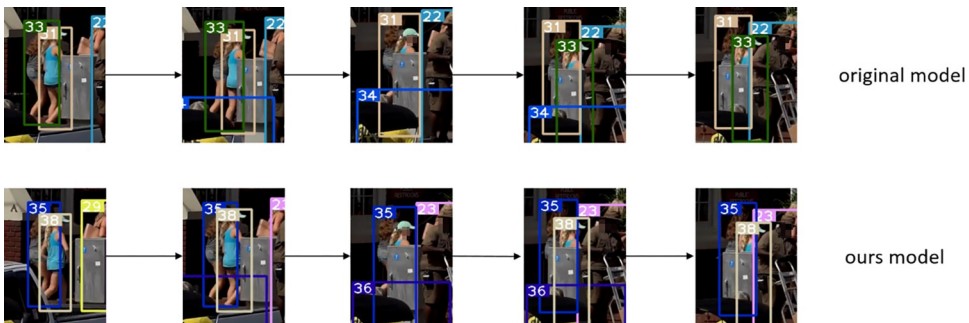

**Fig 15. Partial occlusion experiment results of the target.** Images republished from Vidsplay.com [21] under a CC BY license, with permission, original copyright [2023].

enhancement methods will be considered to make our model applicable to low-resolution datasets.

**MOT17.** The MOT17 dataset contains seven different sequences in both the training and testing sets. It has a higher average target density, and frequent occlusions occur during target motion. In the ablation experiment (Section 4.1), our proposed model and correlation strategy can reduce the number of identity switches. In the comparative experiment (Section 4.2), our algorithm achieves higher accuracy and fewer identity switches compared to recent good algorithms.

Figs 7–9, 12 demonstrate the tracking results of the correlation strategy in this paper when dealing with scenarios of fewer detection frames and more disappearing frames. Additionally, we present selected tracking results for occluded targets, such as in Figs 14 and 15. Our proposed algorithm can correctly assign the correct IDs even in cases of complete occlusion (ID 36) and partial occlusion (ID 31,38), effectively reducing the number of switches during object tracking.

## 5. Conclusion

In this paper, we propose a new network feature extraction module that is more accurate in extracting features. It has a high model robustness and a strong generalization ability. To address the occlusion problem in multi-object tracking, an occlusion-resistant association strategy is designed that uses appropriate algorithms to predict trajectories and makes reasonable use of high-value predicted boxes. The experiments show that our proposed optimization algorithm achieves excellent performance on various indicators of the MOT dataset. The more accurate the detection model is, the better the optimization algorithm in this paper performs.

## Acknowledgments

The pictures of pedestrians used in the manuscript are all taken from https://www.vidsplay.com, datasets are shared, copy and redistribute material in any medium or format.

## Author Contributions

**Conceptualization:** Dongming Li.

**Methodology:** Lijuan Zhang, Gongcheng Ding.

**Project administration:** Yutong Jiang.

**Validation:** Dongming Li.

**Visualization:** Dongming Li.

**Writing – original draft:** Lijuan Zhang, Gongcheng Ding.

**Writing – review & editing:** Guanhang Li, Zhiyi Li.

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
