## [Decision Letter · Decision Letter 0]

23 May 2023

PONE-D-23-09646An Anti-Occlusion Optimization Algorithm for Multiple Pedestrian TrackingPLOS ONE

Dear Dr. Jiang,

Thank you for submitting your manuscript to PLOS ONE. After careful consideration, we feel that it has merit but does not fully meet PLOS ONE’s publication criteria as it currently stands. Therefore, we invite you to submit a revised version of the manuscript that addresses the points raised during the review process. Please submit your revised manuscript by Jul 07 2023 11:59PM. If you will need more time than this to complete your revisions, please reply to this message or contact the journal office at plosone@plos.org. Please include the following items when submitting your revised manuscript:A rebuttal letter that responds to each point raised by the academic editor and reviewer(s). You should upload this letter as a separate file labeled 'Response to Reviewers'.A marked-up copy of your manuscript that highlights changes made to the original version. You should upload this as a separate file labeled 'Revised Manuscript with Track Changes'.An unmarked version of your revised paper without tracked changes. You should upload this as a separate file labeled 'Manuscript'.

We look forward to receiving your revised manuscript.

Kind regards,

Gulistan Raja

Academic Editor

PLOS ONE

Journal Requirements:

4. We note that Figures 3,4,5,7,8,9 and 11 includes an image of a participant in the study. 

Additional Editor Comments:

Reviewer 1 and Reviewer 2 were of the view that manuscript describes a technically sound piece of work and recommended major revision. They have made certain observations/suggestions/comments to further improve your work.After considering comments of both reviewers, the my decision is "major revision". Please incorporate all the comments/suggestions/observations made by both reviewers. **Additional Editor Note to Authors:** I have noted that reviewers have asked for more context in the literature review, and suggested specific papers to be cited. While you may take on-board their suggested papers if you feel that they are relevant for your manuscript, or just take on-board the general suggestion for providing some more context in the literature review, there is no requirement from the journal to cite these papers

Reviewers' comments:

Reviewer's Responses to Questions

**Comments to the Author**

1. Is the manuscript technically sound, and do the data support the conclusions?

Reviewer #1: Yes

Reviewer #2: Yes

2. Has the statistical analysis been performed appropriately and rigorously? 

Reviewer #1: Yes

Reviewer #2: Yes

3. Have the authors made all data underlying the findings in their manuscript fully available?

Reviewer #1: Yes

Reviewer #2: Yes

4. Is the manuscript presented in an intelligible fashion and written in standard English?

Reviewer #1: Yes

Reviewer #2: Yes

5. Review Comments to the Author

Reviewer #1: (1)Some figures, such as Figs. 1, 2, are unclear. The details of the blocks should be clearly defined and explained directly in the figures even they have been explained in text. Please provide the information as much as possible, then the readers can get the details from each separate figure easily even without the text. What do the circles mean? What do the colors of the three feature boxes mean?

(2)The characters in Fig. 6 are too small. Please check it and revise it. What does the circle (with x) mean?

(3)Please check the manuscript carefully to remove the typos, improve the language and format.

E.g.

-Wes uses

...

(4)The review of the related works and comparison experiments can be more sufficient. Please carefully read, cite and compare (if applicable) the following papers that are SOTA tracking. [-SiamCAM: A real-time Siamese network for object tracking with compensating attention mechanism; -Robust visual tracking with occlusion judgment and re-detection; -A scale adaptive object tracking algorithm with occlusion detection;- Double-channel object tracking with position deviation suppression] If the authors cannot employ these methods or compare their method with these methods, at least they could introduce/mention these novel technologies in related sections to improve the quality of the survey.

(5)Please provide and label the reference indices of the compared methods in the figures and tables, such as Tables 1~6, and then the readers can judge whether the compared methods are SOTA.

(6)Why are the compared methods different in the tables and figures? E.g. Tables 7 and 8. All the comparisons should be fair, objective and comprehensive rather than biased, subjective and selective.

Reviewer #2: 1. English writing needs further improvement. The structure of this paper needs to be optimized and the METHOD section (current sections 2 and 3) should be highlighted. At present, it is difficult for readers to find the author's scientific discovery or technological innovation.

2. In Fig. 1, “Cycle-Attention” module has only one input, but in Fig. 2, two inputs appear. Additionally, it is better to draw the output arrow in Fig. 2. In the figures or text, there is no description of the internal structure of the “Self-Attention” module in Fig. 1.

3. Please indicate which attention mechanism or its internal structure is used for the blue attention module on the right in Fig. 2.

4. How is the “cycle shift” operation done in Fig. 2 and what is its purpose? How to construct a large number of negative samples through cycle shift in the “Cyclic-Attention” module? It's better to introduce it. Will generating these negative samples and their subsequent processing introduce a lot of computational overhead? Has the author proposed methods to improve computational efficiency?

5. Figs. 1 and 2 should provide legends for ○+ and ○× operations. What does the ○× symbol in Fig. 6 mean? What does “confirm d” in Fig. 6 mean?

6. Please unify the use of the term “equation”. In Equations 10 and 11, “formula” is used, but all other equations use “equation”.

7. More in-deep analyses, discussion and comparison are needed in Experiments. Competitive results with recent works are needed.

8. Overall, the literature review should be improved. The authors need to review and discuss visual object tracking (For example, DOI: 10.1016/j.asoc.2022.108485; 10.1016/j.compeleceng.2022.107730; 10.1007/s12652-020-02572-0; 10.1109/LSP.2023.3238277 ) and detection (DOI: 10.22967/HCIS.2022.12.023; 10.3233/AIS-220038; 10.1007/s11554-022-01252-w) algorithms. It is suggested that the related papers should be cited.

6. PLOS authors have the option to publish the peer review history of their article (what does this mean?). If published, this will include your full peer review and any attached files.

Reviewer #1: No

Reviewer #2: No

---

## [Author Response · Author response to Decision Letter 0]

25 Jul 2023

Thank you to the editors of this journal for their responsibility towards this manuscript, and thank you to the reviewers for their valuable comments during the review process.

We have made the necessary format modifications based on the formatting guidelines provided by your journal and updated the ORCID iD associated with our account. We would like to express our gratitude for handling our manuscript. We have individually addressed the review comments provided by each reviewer and have uploaded our responses in the system. We sincerely appreciate the time and effort spent by the reviewers in reviewing our manuscript.

---

## [Decision Letter · Decision Letter 1]

1 Sep 2023

An Anti-Occlusion Optimization Algorithm for Multiple Pedestrian Tracking

PONE-D-23-09646R1

Dear Dr. Jiang,

We’re pleased to inform you that your manuscript has been judged scientifically suitable for publication and will be formally accepted for publication once it meets all outstanding technical requirements.

Kind regards,

Gulistan Raja

Academic Editor

PLOS ONE

Additional Editor Comments (optional):

Reviewers' comments:

Reviewer's Responses to Questions

**Comments to the Author**

1. If the authors have adequately addressed your comments raised in a previous round of review and you feel that this manuscript is now acceptable for publication, you may indicate that here to bypass the “Comments to the Author” section, enter your conflict of interest statement in the “Confidential to Editor” section, and submit your "Accept" recommendation.

Reviewer #1: All comments have been addressed

Reviewer #2: All comments have been addressed

2. Is the manuscript technically sound, and do the data support the conclusions?

Reviewer #1: Yes

Reviewer #2: Yes

3. Has the statistical analysis been performed appropriately and rigorously? 

Reviewer #1: Yes

Reviewer #2: N/A

4. Have the authors made all data underlying the findings in their manuscript fully available?

Reviewer #1: Yes

Reviewer #2: Yes

5. Is the manuscript presented in an intelligible fashion and written in standard English?

Reviewer #1: Yes

Reviewer #2: Yes

6. Review Comments to the Author

Reviewer #1: The authors have adequately addressed my comments raised in a previous round of review.

I feel that this manuscript can be accepted now.

Reviewer #2: The article has been revised according to the review comments and can be published without further modification.

7. PLOS authors have the option to publish the peer review history of their article (what does this mean?). If published, this will include your full peer review and any attached files.

Reviewer #1: No

Reviewer #2: **Yes: **Jianming Zhang

---

## [Editor Report · Acceptance letter]

25 Sep 2023

PONE-D-23-09646R1 

An anti-occlusion optimization algorithm for multiple pedestrian tracking 

Dear Dr. Jiang:

I'm pleased to inform you that your manuscript has been deemed suitable for publication in PLOS ONE. Congratulations! Your manuscript is now with our production department. 

Kind regards, 

on behalf of

Dr. Gulistan Raja 

Academic Editor

PLOS ONE